# Evolution of Quality of Life in Persons with Parkinson’s Disease: A Prospective Cohort Study

**DOI:** 10.3390/jcm10091824

**Published:** 2021-04-22

**Authors:** Eduardo Candel-Parra, María Pilar Córcoles-Jiménez, Victoria Delicado-Useros, Antonio Hernández-Martínez, Milagros Molina-Alarcón

**Affiliations:** 1Department of Nursing, Physiotherapy and Occupational Therapy, Faculty of Nursing, University of Castilla-La Mancha, Av. de España, 02001 Albacete, Spain; Eduardo.Candel@uclm.es (E.C.-P.); pilar.corcoles@uclm.es (M.P.C.-J.); victoria.delicado@uclm.es (V.D.-U.); 2Department of Nursing, Physiotherapy and Occupational Therapy, Faculty of Nursing, University of Castilla-La Mancha, 13071 Ciudad Real, Spain; 3Instituto de Investigación en Discapacidades Neurológicas (IDINE), University of Castilla-La Mancha, Av. de España, 02001 Albacete, Spain

**Keywords:** Parkinson’s disease, quality of life, PDQ 39, motor symptoms, no-motor symptoms

## Abstract

Parkinson’s disease (PD) is a chronic neurodegenerative disorder that results in important functional symptoms, altered mood, and deterioration in quality of life (QoL). This study aimed to determine the evolution of the QoL in persons with PD in the Albacete health district over a two-year period and identify associated sociodemographic, clinical, and socio-health characteristics. A cohort study was conducted of patients at different stages of PD in the Albacete health district. Calculated sample size: 155 patients. Instruments: A purpose-designed questionnaire for data collection and the “Parkinson Disease Questionnaire” (PDQ-39), which measures 8 dimensions and a global index where a higher score indicates worse quality of life. Three measurements were made: baseline, one year, two years. A descriptive and bivariate analysis was conducted. Ethical aspects: informed consent, anonymized data. Results: Mean age 69.51 (standard deviation, SD 8.73) years, 60% male, 75.5% married, and 85.5% lived with family. The most frequent motor symptoms were slow movement (86.23%), postural instability (55.5%), tremor (45.5%), and dyskinesia (24.6%). Among the non-motor symptoms were fatigue (66.2%), pain, daytime somnolence, constipation, and apathy, with approximately 50% each. The mean QoL score at baseline was 27.47 (SD 16.14); 95% CI (confidence interval) 24.91–30.03. At two years, global QoL had slightly worsened (28.3; SD 17.26; 95% CI 25.41–31.18), with a statistically significant worsening in mobility, activities of daily living, and communication, whereas social support improved.

## 1. Introduction

Parkinson’s disease (PD) is a chronic progressive and invalidating neurodegenerative disease requiring important lifestyle modifications. The mean age at diagnosis is between 60 and 65 years [1]. From a clinical point of view, two large groups of disorders are distinguished. One refers to the motor symptoms (MS), with poverty or slowness of movements (akinesia), increased muscle tone or rigidity, and abnormal involuntary movements (dyskinesias). The association of tremor, rigidity, akinesia-bradykinesia, and alterations in postural reflexes constitute what is called “parkinsonian syndrome.” The other disorder refers to the non-motor symptoms (NMS), with features that indicate the existence of autonomic (constipation, hyperhidrosis), sensation (paresthesia, pain), and mental dysfunction (depression, dementia). PD is the second most frequent neurodegenerative disease in our region, following Alzheimer’s disease, causing more than 4600 deaths annually (11th place in causes of death: 1.1% of total), and its death rate has been increasing over the last two decades [2]. PD has a substantial impact from the moment of diagnosis on the health-related quality of life (HRQoL) of patients and their family caregivers. Another important aspect is that half of the patients suffer from depression at some point during their illness [3].

Cognitive deterioration is very prevalent and has a severe, negative effect on health and on perceived HRQoL. Various biomarkers exist that are associated with cognitive decline, including clinical, neuropathological, genetic, and neuroimaging. It originates from a progressive loss of dopaminergic neurons in grey matter and the presence of inclusion bodies called Lewy bodies. This neuronal death leads to a deficit of dopamine in the striatal endings that transmit information for the correct control of movements. As in most neurodegenerative diseases, the cause is unknown; although, it is probably multifactorial, including factors related to the person’s biological characteristics, genetics, environment, and aging [3,4,5].

In Spain, the most recent statistics indicate that between 70,000 and 100,000 people are affected. The estimated incidence in Spain is between 20 to 25 new cases per year per 100,000 inhabitants. Hospital clinical records note a world prevalence of between 100 and 300 cases for every 100,000 inhabitants [6,7]. PD is more common in males than in females, and the cause of this discrepancy is unknown.

The typical clinical MS at presentation are tremor, rigidity, and slow movement. More than 40% of those affected have postural and action tremor, and between 10 and 25% do not experience tremor. Tremor is the symptom that most affects HRQoL, despite slowness and rigidity being the symptoms that limit normal functionality. Rigidity is responsible for the muscle pain experienced by patients with PD. Nevertheless, the most severe motor problem is akinesia, with different degrees of severity in movement and changes in position [3,6].

NMS are common among patients with early PD and represent an important cause of worse HRQoL, especially depression and sleep disturbances, which are associated with reduced wellbeing. Mood and cognitive disorders affect 80.6% of patients [8,9]. Behavioral, psychological, and/or mental problems are often not evident initially, but they become a serious problem with time. Among the most common neuropsychiatric characteristics in patients with PD, apathy, anxiety, and depression stand out. Depression is one of the most important factors that affect HRQoL in this patient group [10]. Apathy is not usually considered as a specific neuropsychiatric conditioner, but rather as a sign related to depression or cognitive impairment, and it may be ignored due to its overlap with motor disability and hypokinesia [11]. The mood and affective disorders present with depression and anxiety, sometimes, before diagnosis can include obsessive and repetitive conducts, generalized anxiety disorders, and panic or anxiety attacks during off periods. Antidepressants improve anxiety and depression at some point in 40–50% of patients with PD. Depression (associated or not with anxiety) is a neuropsychiatric disorder that has shown the greatest effect on worse HRQoL in these patients [9,12]. These results are directly associated with HRQoL, independently of the severity of MS. Prognostic variables include depressive symptoms, insomnia, and a low degree of independence in relation to the severity of the disease. Half of patients complain of a lack of energy related to depressive symptoms [13,14].

The early identification and management of neuropsychiatric symptoms are essential for preserving HRQoL [15]. The most frequent psychosocial difficulties are related to cognitive development, motor development, and alterations in mood, such as depression and anxiety. These mean lower HRQoL and greater disability [16]. The presence of NMS is also associated with freezing of gait [14] and memory problems [17] and has a negative impact on the HRQoL as measured by the Parkinson Disease Questionnaire PDQ-39 [18].

The Hoehn and Yahr (HY) scale is utilized to classify the evolution and progression of this disease. This scale was the first used to evaluate this disease and is based on motor symptoms, identifying five stages from more minor to worse disability [19].

Curative treatment still does not exist for this disease; therefore, therapy focuses on alleviating symptoms, delaying motor complications, and attempting to prolong autonomy for as long as possible. Treatment may be pharmacological, surgical, and/or involve rehabilitation, and all these can be combined. Among pharmacological treatments, levodopa is used as well as other dopaminergic agonist drugs. Surgical techniques may facilitate improvement, in some cases, of the symptoms of this disease, and evidence exists for improvement in HRQoL as well. Finally, physiotherapy and rehabilitation improve mobility and performance of activities of daily living (ADLs) [20,21].

The present study aimed to determine the evolution of the quality of life of persons with PD in the Health Area of Albacete over the course of two years and identify the sociodemographic, clinical, and socio-health characteristics of this population.

## 2. Materials and Methods

Design: Analytical observational longitudinal prospective cohort study.

Study population: Patients diagnosed with PD and treated in the Department of Movement Disorder (UTM in Spanish) of the Neurology Service of the Integrated Management Area (GAI in Spanish) of Albacete between 2015 and 2018.

Inclusion criteria: All patients diagnosed with PD treated at the UTM that voluntarily agreed to participate in this study. Patients with cognitive deterioration, as documented in their clinical history, those that were not able to be contacted via post or phone, and those that did not understand Spanish were excluded.

Sample size: For an estimated prevalence of PD in the general population of 187 cases per 100,000 inhabitants [8] and with a population in the health coverage area of Albacete in 2015 of 414,892 inhabitants, around 776 persons could be expected to be diagnosed. If the mean score and standard deviation of the HRQoL were similar to that obtained in the Rahman study (mean 32.4; standard deviation, SD = 16.3) [22], which was conducted with patients with PD who were treated in the Movement Disorder Service of the University College of London and used the PDQ-39 instrument, for a confidence level of 95% and a precision of ±2.5, the estimated sample size was 135 subjects (EpiDatv3.1, Service of Epidemiology of the Dirección Xeral de Saúde Pública, Santiago de Compostela, Spain). This sample size was increased by 15%, considering a possible dropout or loss to follow-up, resulting in a total sample of 155 subjects. The participants were selected consecutively until the estimated sample size was reached, from January 2015 to December 2016.

The variables studied were as follows: sociodemographic variables (age, sex, civil status, living situation, employment situation, education level), clinical variables (stages of HY, duration of PD, measured from the date of diagnosis to the date of inclusion in the study), other medical history, deep brain stimulation (DBS), presence of MS and NMS, apathy, depression, pharmacological treatment, and non-pharmacological treatment. In addition, socio-health variables were studied, including the presence of a caregiver, dependence assessment, degree of disability, assistance from long-term care systems, and mortality. The principal variable was HRQoL, which was measured at three timepoints: baseline (T0), one year (T1), and at two years (T2).

Measuring instruments:Data collection questionnaire: Purpose-designed.“Parkinson’s Disease Questionnaire” (PDQ-39) [18] that measures HRQoL and contains 39 items, which cover 8 dimensions. These dimensions are mobility, activities of daily living (ADL), emotional wellbeing, social support, stigma, communication, cognitive state, and pain. Each item is scored on a scale of 5 (from 0 = never to 4 = always or unable to perform). The results are calculated as a percentage, totaling the scores of each dimension’s items, multiplying by 100, and dividing by the maximum score of the dimension. A higher score indicates worse HRQoL. A global index can be obtained by calculating the mean of each dimension’s scores (PDQ Summary Index, PDQ-39 SI) [18] that summarizes the results of each scale. This questionnaire has been used in clinical trials, in which the variations in the different dimensions were in agreement with clinical assessments made using the usual PD scales, which supports its adequate sensitivity to clinical changes in patients. Among the questionnaire requirements is the need to be filled out via a personal interview at the time.

Procedures: First, a pilot study was conducted before the data collection phase with 10 patients to evaluate the purpose-made questionnaire’s validity for data collection. An information sheet outlining the study objectives was created, and written informed consent was requested from those who wished to participate.

Data collection was conducted in the UTM, after consent had been obtained, by conducting an interview with the patients and primary caregivers and reviewing the clinical history. The questionnaires were repeated at one and two years via telephone interview. When patients were unable to respond via phone, an in-person interview was conducted in their home.

The statistical analysis was conducted using the program IBM SPSS v.24. (IBM, Armonk, NY, USA). A descriptive statistical analysis was conducted for each of the variables. For qualitative variables, the absolute and relative frequencies were calculated, and for quantitative variables the mean and standard deviation (SD) were calculated. The Kolmogorov-Smirnov and Levene tests were used to analyze the normality and homoscedasticity of the distributions. Confidence intervals (CI) were set at 95%. A bivariate analysis was conducted, comparing HRQoL scores (PDQ-39 SI and its separate dimensions) in each successive observation timepoint using the repeated-measures ANOVA (analysis of variance). The statistical significance level was set at *p* < 0.05.

Ethical aspects: This study was conducted in accordance with the principles of the Declaration of Helsinki with regard to studies involving human subjects, and also in line with Law 14/2007 for biomedical research. The principles of confidentiality and anonymity in the treatment of the data and presentation of the results were respected at all times, in line with legislation (EU) 2016/679 of the European Parliament and Council on 27 April 2016, concerning the protection of natural persons in terms of the processing of personal data and free movement of such data. This study was approved by the Clinical Research Ethics Committee (CEIC) for the Albacete Health Area (Report 03/11) and the Clinical Research Commission of the GAI of Albacete.

## 3. Results

A final 155 subjects with valid data were included. At T1, 148 subjects were still followed, and 141 subjects at T2. All patients lost to follow-up were due to death. Descriptive data refer to baseline characteristics (T0). The study cohort’s mean age was 69.51 (SD 8.63) years, ranging from 43 years to 89 years. The percentage of males was 59.4% (92).

The principal results that describe the study population are shown in Table 1. In terms of family situation, 75.5% (117) of the patients were married, and 16.1% (25) were widowed. The majority, 85.5% (133), of the patients lived with their families, and 4.5% (7) were institutionalized. The majority were retired, 73.5% (114), and 82.6% had primary level education.

In terms of clinical characteristics and years of PD evolution, patients had a mean duration of PD of 9.71 years (SD = 6.46). The duration of PD was less than 5 years in 29% (45), 6 to 10 years in 36.1% (56), 11 to 15 years in 18.8% (29), 16 to 20 years in 11.6% (18), and more than 20 years in 4.2% (7). Twenty-nine (18.7%) of the patients had been diagnosed before 50 years of age. Among the most frequent comorbidities were diabetes, hypertension with 12.3% (19) each, arthritis 7.1% (11), prostatism 10.87% (17), and cardiopathies 5.2% (8).

MS were present in the different stages of PD, and the most frequent were slow movement in 86.23% (134), postural instability in 55.5% (86), difficulty turning in 49.1% (76), tremor in 45.5% (71), and dyskinesias in 24.6% (38).

Among the NMS, fatigue was present in 66.2% (96), pain in 50.7% (79), daytime somnolence in 49.6% (77), urinary incontinence 50.2% (78), depression-apathy in 45.4% (70), constipation in 54.1% (84), insomnia in 36.5% (57), and dysphagia in 31.2% (48).

The mean of MS present in patients was 2.59 (SD 1.3), NMS was 4.2 (SD 2.9), and both MS and NMS present at the same time was 6.8 (SD 3.2).

In terms of the clinical progression of PD, at the start of the study about 67% (114) were in stages 1 and 2 according to the HY classification. Of these patients, 62.7% (93) remained in the same stages at one year (T1) and 53.2% (75) at two years (T2). For stages 3, 4, and 5, around 33% (46) of participants were classified as being in these stages at the start of the study, increasing to 37.2% (55) at one year, and 46.8% (66) at two years.

In terms of pharmacological treatment, the majority of the patients took rasagiline (61.35, 95 patients), either alone or in combination with levodopa, and the combination levodopa + carbidopa + entacapona (54.2%, 84 patients). In addition, 37.4% (58) took antidepressants. The percentages of patients that were taking more than 4 drugs were 41.3% (64) (T0), 44.6% (66) at T1, and 45.4% (64) at T2.

In terms of non-pharmacological treatment and health recommendations, 45.2% (70) of patients reported going for walks, 32.9% (51) did physiotherapy, 21.9% (34) did occupational therapy, and 7.7% (12) SLT. Psychological therapy was used by a small percentage of the sample, 3.2% (5).

At the start of the study, 40.0% (62) of patients had a caregiver, increasing to 50% (73) at both one and two years. Of the patients with a caregiver, the caregiver was most often the wife in 41.9% (26), followed by a daughter in 11.6% (7).

Concerning HRQoL, the mean global score (PDQ-39 SI) and the different dimensions, along with their evolution at one and two years, are detailed in Table 2.

The differences in the different HRQoL dimensions scores over the two years are shown in Table 3.

## 4. Discussion

Evaluation of the global HRQoL in the patients included in our study showed that the majority presented scores indicating that their quality of life was partially affected but without major problems. The score obtained in the sample using the PDQ-39 SI at the time of enrollment (T0) was 27.47, at one year (T1) 28.08, and at two years (T2) 28.30.

When analyzing the evolution of HRQoL over the two years of follow-up we did not observe any significant worsening in global HRQoL. The mean score on the PDQ-39 SI was slightly lower than that of other studies, suggesting our patients had a better HRQoL than in the study by Rahman et al., 2008, carried out on a population with PD attended by the Movement Disorders Centre of the University College London, in which they had a mean HRQoL score of 32.4, measured with PDQ-39 [22], and in the study Mínguez et al., 2015, in Albacete, in which they obtained a score of 33.47 in the PDQ-39 SI [23]. The studies by Ayala et al., 2017, and Llagostera-Reverter et al., 2019, coincide with our study as they also found a small change in global HRQoL after a three-year follow-up and found that the progression of the disease depended more on the baseline characteristics of the individual [24,25].

The HRQoL dimensions most affected were mobility, emotional wellbeing, ADL, pain, and cognitive status. The dimensions of the PDQ-39 questionnaire with the worst score, and therefore most affected in T0, from highest to lowest, were those related to emotional wellbeing with a score of 36.23, mobility (36.08), pain (32.36), ADL (32.04), and cognitive status (29.27). The least affected were stigma, communication, and social support, which suggests that these patients maintained family and social stability, and that stigmatization due to PD was not relevant in our setting. Our results are in agreement with other studies that mobility is the most affected dimension [3,26].

Strupp et al., 2018, found the most common dimensions to be severely affected were mobility (34.9%), coordination problems, speech problems, and limited daily activities [27].

In the pain dimension, patients reported the presence of painful muscle cramps and spasms, unpleasant sensations of heat and cold, as well as joint pain.

The importance of this dimension was revealed in the study by Buhmann et al., 2017, which stated that pain was present in 40% of cases; of these, 91.1% suffered from chronic pain, and this was diagnosed in only 22.3% [28]. Choi SM et al., 2017, noted that patients with pain, regardless of the pain subtype, had a worse PDQ-39 score than those without pain. Pain along with depression, decreased ability to perform ADLs, and early onset of PD symptoms were associated with poor HRQoL [29]. Given that pain is one of the dimensions of HRQoL that shows the greatest deterioration, interventions aimed at its prevention and relief should be considered a priority and relevant in patients with PD, as adequate pain management would impact an improvement in HRQL.

The emotional wellbeing dimension appears to remain fairly stable, although the patients in our study reported feelings of depression, loneliness, isolation, and a desire to cry. Knie et al., 2011, reported that early-onset was a risk factor for lower emotional wellbeing. The risk of depression and excessive daytime sleepiness were elevated in PD patients compared with controls [30]. This indicates the need for accurate diagnosis and treatment of depression in early-onset patients, which could improve their HRQoL [30]. Buhmann et al., 2017, found that high levels of pain were associated with higher scores for depression and anxiety and lower HRQoL. Their results showed that pain in PD is frequent, complex, and impairs HRQoL, but is underdiagnosed and untreated [28].

Regarding ADLs, most of the patients reported little difficulty in personal hygiene, dressing themselves, buttoning clothing, or tying shoelaces. They maintained the ability to cut food and hold a glass without spilling the contents but had trouble writing clearly. Patients highly value these activities, and their deterioration would affect HRQoL. As stated in the study by Choi et al., 2017, it was observed that pain and depression, difficulties with ADL, and an earlier age of onset of PD symptoms were associated with poor HRQoL [29].

In terms of global HRQoL measures, we did not find statistically significant changes in the evolution of the PDQ-SI over two years, showing only a slight increase in the mean values. However, statistically significant differences were observed when comparing the evolution of this measure’s different dimensions (PDQ-39). Specifically, mobility, ADLs, and communication worsened (increasing value), while social support improved. These findings are in line with those of Martínez-Martín et al., 2017, who concluded that the deterioration of HRQoL progressed as PD progressed [31].

Regarding sociodemographic characteristics, similarities were observed with the study by Llagostera et al., 2019, in terms of the high mean age, male predominance, married marital status, and family living [25].

Considering the clinical situation, the patients studied mainly were in stages I, II, and III of HY and had been living with the disease for a mean of 9.71 years (SD ± 6.46), which is similar to other studies that had a mean time of evolution of PD of 8.1 (SD ± 5.2 years) [25,32,33]. We also found diabetes, arterial hypertension, osteoarthritis, prostatism, and heart disease were the most frequent comorbidities, consistent with another study in which comorbidity varied, and higher percentages of diabetes, arterial hypertension, and prostatism were found [34].

Considering the study’s limitations, we should note the selection bias of the sample. Subjects were recruited among those treated at the UTM, a referral center for patients from other provinces of the autonomous community. This recruitment pool could have resulted in better HRQoL scores, as patients referred from other provinces could find themselves in a more complex clinical situation and therefore unable to be clinically managed in their hospital of origin. The sociodemographic and socio-health variables were collected by self-reporting of patients in our sample; therefore, there could be some memory bias or concealment of sensitive data. Additionally, the recruitment system through face-to-face patient consultation in this unit may have reduced the inclusion of patients in the most advanced stages because those with worse mobility conditions may experience more difficultly traveling to the hospital consultation.

## 5. Conclusions

The majority of the patients with PD at the time of recruitment were in the early stages of the disease and had comorbidities, with diabetes, hypertension, and prostatism being the most frequent. The most common MS and NMS included slow movement, postural instability, difficulty turning, tremor, dyskinesias, fatigue, pain, daytime somnolence, urinary incontinence, apathy-depression, constipation, and dysphagia. All the patients were receiving pharmacological treatment. The most common non-pharmacological treatments used were walking, physical therapy, speech therapy, and occupational therapy.

The HRQoL of patients treated at the UTM in the Albacete Management of Integrated Care was similar to that found in patients studied within the European context, although the global HRQoL values were slightly better. HRQoL did not significantly worsen over the two years of study follow-up; therefore, we consider there is no relevant deterioration in the short and medium term. The dimensions of HRQoL with the worst score, and therefore with the greatest impact, were those referring to emotional wellbeing, mobility, pain, ADL, and cognitive status. On the contrary, the stigma, communication, and social support dimensions were the least affected. Over the two-year study period, significant deterioration was observed in the mobility, ADL, and communication dimensions; in contrast, the social support dimension improved during this period. Emotional aspects were the most affected at the beginning of the disease, with mobility, pain, and ADLs being affected later.

## Figures and Tables

**Table 1 jcm-10-01824-t001:** Sample characteristics.

Variable	% (*n*)
**Mean Age (SD)**	69.51 (8.63)
**95% CI**	68.14–70.88
**Sex**	
Male	59.4 (92)
Female	40.6 (63)
**Living situation**	
With family	85.8 (133)
Lives alone	9.7 (15)
In a residence	4.5 (7)
**Civil status**	
Married	75.5 (117)
Widowed	16.1 (25)
Divorced	1.9 (3)
**Employment situation**	
Retired	73.5 (114)
Housework	19.4 (30)
Actively employed	5.2 (8)
**Education**	
No education	0.6 (1)
Primary level	82.6 (128)
Secondary level	12.3 (19)
University level	4.5 (7)

SD, standard deviation, CI, confidence interval.

**Table 2 jcm-10-01824-t002:** Evolution of global HRQoL and each dimension of the PDQ-39 over the two years of follow-up (T0-T1-T2).

	T0 *n* = 155 Mean (SD) 95% CI	T1 *n* = 148 Mean (SD) 95% CI	T2 *n* = 141 Mean (SD) 95% CI
PDQ 39-SI	27.47 (16.14) 24.91–30.03	28.08 (16.42) 25.42–30.75	28.46 (17.26) 25.41–31.18
**PDQ-39 DIMENSIONS**			
Mobility	36.08 (30.18) 29.99–39.97	39.08 (30.99) 34.05–44.12	42.64 (31.36) 37.12–47.58
ADL	32.04 (27.71) 25.82–34.76	33.81 (29.72) 28.98–38.63	35.57 (29.8) 30.35–40.3
Emotional Wellbeing	36.23 (24.03) 32.23–40.05	35.58 (24.85) 31.54–39.62	37.38 (27.11) 32.59–41.62
Stigma	19.23 (24.55) 15.33–23.13	15.66 (22.75) 11.97–19.36	14.53 (23.07) 9.97–17.97
Social Support	18.49 (18.54) 15.55–21.43	17.39 (15.36) 14.9–19.89	8.51 (14.9) 6.02–11
Cognitive State	29.27 (20.95) 25.94–32.59	29.26 (21.1) 25.83–32.69	30 (22.79) 26.26–33.9
Communication	16.07 (19.99) 12.9–19.24	20.15 (22.79) 16.45–23.86	26.06 (27.47) 21.48–30.63
Pain	32.36 (23.02) 28.71–36.01	33.72 (22.42) 30.08–37.37	32.97 (23.37) 29.08–36.87

Purpose-made questionnaire. T0 is considered the baseline observation of the sample participants, T1 is the second observation point at 1 year of evolution, and T2 is the third observation at two years of evolution. The results of each dimension are shown as mean, SD, and 95% CI. HRQoL, health-related quality of life; PDQ-39 Parkinson Disease Questionnaire; PDQ 39-SI, PDQ-39 Summary Index; ADL, activities of daily living.

**Table 3 jcm-10-01824-t003:** Evolution of HRQoL considering the total scores of each of the PDQ-39 dimensions for the patients that completed follow-up.

Comparisons T0–T1 and T0–T2 (*n* = 141)					
**Mobility Dimension**					
**Period**	**Mean (SD)**	**Mean Difference (SD)**		**95% CI**	***p*-Value**
**Initial (T0)**	34.98 (29.95)	**T0–T1**	−3.67 (1.52)	−7.37, 0.03	0.053
**1st year (T1)**	38.65 (31.02)	**T1–T2**	−3.98 (1.16)	−6.8, −1.17	0.002 *
**2nd year (T2)**	42.64 (31.36)	**T0–T2**	−7.66 (1.75)	−11.90, 3.41	0.001 *
**AVD Dimension**					
**Period**	**Mean (SD)**	**Mean Difference (SD)**		**95% CI**	***p*-Value**
**Initial (T0)**	30.46 (26.73)	**T0–T1**	−3.19 (1.37)	−6.51, 0.13	0.065
**1st year (T1)**	33.65 (29.43)	**T1–T2**	−1.92 (1.27)	−5.00, 1.16	0.402
**2nd year (T2)**	35.57 (29.8)	**T0–T2**	−5.11 (1.64)	−9.09, 1.13	0.007 *
**Stigma Dimension**					
**Period**	**Mean (SD)**	**Mean Difference (SD)**		**95% CI**	***p*-Value**
**Initial (T0)**	18.35 (24.38)	**T0–T1**	−3.01 (1.41)	−0.40, 6.43	0.103
**1st year (T1)**	15.33 (22.75)	**T1–T2**	−0.79 (1.13)	−1.95, 3.54	1.000
**2nd year (T2)**	14.53 (23.07)	**T0–T2**	3.81 (1.61)	−0.10, 7.72	0.059
**Communication Dimension**					
**Period**	**Mean (SD)**	**Mean Difference (SD)**		**95% CI**	***p*-Value**
**Initial (T0)**	14.65 (18.39)	**T0–T1**	−5.49 (1.41)	−8.69, −2.29	0.001 *
**1st year (T1)**	20.15 (22.14)	**T1–T2**	−5.91 (1.53)	−9.63, −2.18	0.001 *
**2nd year (T2)**	26.06 (27.47)	**T0–T2**	−11.4 (1.81)	−15.81, −6.99	0.001 *
**Social Support Dimension**					
**Period**	**Mean (SD)**	**Mean Difference (SD)**		**95% CI**	***p*-Value**
**Initial (T0)**	17.55 (17.41)	**T0–T1**	0.53 (1.4)	−2.87, 3.94	1.000
**1St year (T1)**	17.02 (15.29)	**T1–T2**	−8.51 (1.46)	4.96, 12.05	0.001 *
**2nd year (T2)**	8.51 (14.85)	**T0–T2**	9.04 (1.69)	4.93, 13.15	0.001 *
**Emotional Wellbeing Dimension**					
**Period**	**Mean (SD)**	**Mean Difference (SD)**		**95% CI**	***p*-Value**
**Initial (T0)**	34.92 (23.78)	**T0–T1**	−0.35 (1.58)	−4.2, 3.49	1.000
**1st year (T1)**	35.28 (24.66)	**T1–T2**	−2.09 (1.47)	−5.67, 1.48	0.473
**2nd year (T2)**	37.38 (27.11)	**T0–T2**	−2.45 (1.93)	−7.14, 2.23	0.621
**Cognitive Status Dimension**					
**Period**	**Mean (SD)**	**Mean Difference (SD)**		**95% CI**	***p*-Value**
**Initial (T0)**	27.74 (19.89)	**T0–T1**	−1.24 (1.35)	−4.52, 2.04	1.000
**1st year (T1)**	28.98 (20.77)	**T1–T2**	−1.02 (1.49)	−4.64, 2.60	1.000
**2nd year (T2)**	30 (22.79)	**T0–T2**	−2.26 (1.68)	−6.34, 1.82	0.547
**Pain Dimension**					
**Period**	**Mean (SD)**	**Mean Difference (SD)**		**95% CI**	***p*-Value**
**Initial (T0)**	32.62 (23.04)	**T0–T1**	−1.65 (1.56)	−5.44, 2.13	0.877
**1st year (T1)**	34.27 (22.38)	**T1–T2**	1.3 (1.53)	−2.41, 5.01	1.000
**2nd year (T2)**	32.97 (23.37)	**T0–T2**	−0.35 (1.69)	−4.46, 3.76	1.000

Purpose-made questionnaire. T0 is considered the baseline observation of the sample participants, T1 is the second observation point at 1 year of evolution, and T2 is the third observation at two years of evolution. The results are shown as mean and standard deviation. (*) Indicates statistical significance.

## Data Availability

The data sets generated and/or analyzed during the current study are available from the corresponding author on reasonable request.

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
