# Peer review of "Evolution of Quality of Life in Persons with Parkinson’s Disease: A Prospective Cohort Study"

_jcm, 2021, doi:10.3390/jcm10091824_

Round 1
Reviewer 1 Report
The manuscript by Candel-Parra and colleagues provides an descriptive overview report on PD patients in the Albacete Health district followed by 2 years.
The manuscript is scientifically sound. Statistics was performed appropriately and the text reads well.
Author Response
Thank you for your positive review.
Reviewer 2 Report
the study design and results seem "partial" to me, the patient population "limited", I would expand and diversify the population in order to relate the PDQ-39 at least to a scale for motor symptoms that is more informative than the H&Y and with a scale on non-motor and / or cogitive symptoms. In this version, the scientific relevance seems poor to me and the conclusions of poor interest, although correct.It could be interesting to extend follow up?
Author Response
Thank you for your review and suggestions.
To date, it has not been possible to obtain a large sample size. The study type involves following a cohort of patients with Parkinson’s Disease (PD), over two years, in a specific geographic area. In the Methods, we outline how we conducted the sample selection (lines 129-139) with Epidat 3.1.
The HY scale is the scale most utilized in patients with PD by neurologists. Patients were classified in their clinical records according to this scale. We used the evaluation from the clinical records.
The aim of our study was not to detect MS and NMS in this patient group, rather their influence on HRQoL. The MS and NMS were recorded in their clinical history.
In terms of scientific relevance, our study obtained better results for quality of life than other publications to date. We have highlighted the importance of the burden incurred with the disease for patients and their close network. Hence, stigma and social support is very important in our study, and it would be recommended to consider this aspect in patients with PD.
Reviewer 3 Report
The article "Evolution of quality of life of persons with Parkinson's disease: a prospective cohort study" by Candel-Parra et al. focuses on the evolution of the quality of life of patients with PD.
The study is well-designed. There are some comments though:
The introduction section is too long and should be shortened. On the other hand, this study on the quality of life of patients with PD should address also discuss in the introduction the quality of life or burden of the relatives/caregivers.
There are some linguistic mistakes and the manuscript should undergo grammatical/linguistic editing. Examples: page 3, line 145 – studied in stead of studies and caregiver instead of carer. page 5, line 224 – took in stead of take. Page 6, line 231: caregiver in stead of carer. Page 9, line 322 – "This recruitment pool could have conditioned better HRQoL scores…"
The first paragraph (page 9, line 332-343) of the conclusion is much too long and the paragraph is a repetition of the results. Hence, most of the paragraph is redundant and can be shortened to the most important findings.
Author Response
Thank you for your review and helpful suggestions to improve the quality of the mansucript.
In terms of the introduction, we believe that we can shorten it by cutting lines 110-116 without affecting its content.
In this study, we did not include the quality of life of the family members/caregivers, as the aim was the HRQoL of patients with PD. However, we appreciate the suggestion as it would be very useful for further studies.
The linguistic errors have been corrected, and the manuscript reviewed by a native English medical editor.
In terms of the conclusion, page 9 (lines 332-343), the final version is as follows:
“The majority of the patients with PD at the time of recruitment were in the early stages of the disease and had comorbidities, with diabetes, hypertension, and prostatism being the most frequent. The most common MS and NMS included slow movement, postural instability, difficulty turning, tremor, dyskinesias, fatigue, pain, daytime somnolence, urinary incontinence, apathy-depression, constipation, and dysphagia. All the patients were being treated with pharmacological treatment. The most common non-pharmacological treatments used were walking, physical therapy, speech therapy, and occupational therapy.”
Round 2
Reviewer 2 Report
The work can be accepted in this form after the modifications